# Search Space Synthesis for Parametric Functions

Felix Laarmann [1 2]   Andreas Pauly [1 2]   Sebastian Buschjäger [2]   Andrea Bommert [3]   Jakob Rehof [1 2]

## Abstract

We present a general framework for synthesizing search spaces of parametric functions, along with strategies for traversing these spaces to find optima. We formalize an algebraic theory for the categorical model of parametric functions in finite combinatory logic with predicates (FCLP). Based on a component-oriented synthesis framework for FCLP we automate composition from given components and search for parametric functions. Components are language-agnostic and may be instantiated as any implementation of parametric functions, e.g., as PyTorch modules. A proof-of-concept implementation demonstrates how to represent more specific concepts, such as neural architecture search and hyperparameter optimization, within the framework.

## 1. Introduction

With the proliferation of machine learning (ML) techniques and their success across many application domains, there is a natural interest in understanding the essential structures underlying various classes of machine learning models from an abstract mathematical perspective. In particular, abstract models of classical ML techniques, such as backpropagation, have recently been proposed within category theory (Fong et al., 2019; Cruttwell et al., 2022; Gavranović et al., 2024). Inspired by these developments, we propose to put these abstract models to use by automating the construction of model classes and their implementations using program synthesis techniques (David & Kroening, 2017). We provide a highly general and flexible framework for automated *search space synthesis* for classes of models, focusing on *parametric functions*, along with strategies for traversing them to find optima. A useful characteristic of categorical

models is that they are, by nature, *compositional*. Using a framework based on typed combinatory logic FCLP (finite combinatory logic with predicates (Dudenhefner et al., 2023)) for component-oriented synthesis (Rehof & Vardi, 2014), we can exploit compositional structure to decompose families of complex models into simpler building blocks (components, represented as combinators in combinatory logic), enabling component reuse and simplifying specification. We provide a proof-of-concept implementation that showcases how neural architecture search (NAS) and hyperparameter optimization (HPO) methods arise naturally as implementations within our framework, on the one hand, and how our approach provides the flexibility to synthesize and explore novel (to the best of the authors' knowledge) search spaces for ML models, on the other hand.

**Related Work:** Existing approaches, such as BOHNAS (Schrodi et al., 2023) and Einspace (Ericsson et al., 2024), use extensions of context-free grammars (CFGs) as search spaces. These approaches require users to write the grammars for search spaces, whereas our approach provides a specification language and synthesizes search spaces represented as tree languages, along with search strategies for a given specification. In contrast to manually designed architecture grammars, the proposed approach automatically derives search spaces from typed component specifications while guaranteeing the structural well-formedness of synthesized models. Our approach aims to achieve higher degrees of automation and greater ease of use for the ML community. The FCLP framework (Dudenhefner et al., 2023) can represent solution spaces that go beyond context-free structures by (automatically) representing them as Horn-theories, although the present application is confined to tree languages. Closely related in spirit to ours is the work (Negrinho et al., 2019), which defines a domain-specific language for NAS search spaces over computational graphs and their components. The main differences to our approach are that our search spaces are automatically synthesized from component specifications, and specifications are rooted in a more general categorical model. Moreover, our components are typed so that synthesized solutions are guaranteed to be well-formed with respect to the categorical model. In their ICML 2024 position paper (Gavranović et al., 2024), the authors propose using a structure called the **Para**-construction to represent parametric functions in a

---

[1]Department of Computer Science, TU Dortmund University, Dortmund, Germany [2]Lamarr Institute, Dortmund, Germany [3]Department of Statistics, TU Dortmund University, Dortmund, Germany. Correspondence to: Felix Laarmann <felix.laarmann@udo.edu>.

*Proceedings of the 43$^{rd}$ International Conference on Machine Learning*, Seoul, South Korea. PMLR 306, 2026. Copyright 2026 by the author(s).

generic way. It was proposed that one should search for structures *within* the **Para**-construction to identify categorical datatypes for different ML concepts. This may provide specific categorical models for various ML concepts and enable researchers to reason about them. However, if the primary goal is not to reason about models but to synthesize software that implements them (as in our case), the fact that each of these structures is itself a parametric function (since categories are closed under composition) turns out to be useful, because modern frameworks like PyTorch, TensorFlow, etc., usually provide software components that already implement various ML concepts as parametric functions. A contribution of this paper is to leverage the internal structure of the **Para**-construction *itself* to provide a data structure for (general) parametric functions at the abstraction level of modern ML frameworks. Therefore, it may be viewed as a generalization of the position in (Gavranović et al., 2024) and as a strategy for applying categorical models in modern ML practice. To our knowledge, a specialization of the **Para**-construction to the so-called **Para**(**Lens**)-construction was first implemented in (Cruttwell et al., 2022) to demonstrate its applicability to learning neural networks.

In their ICML 2010 paper (Liang et al., 2010), the authors employ untyped combinatory logic to program learning in an interesting way, which, however, differs significantly from our use of combinatory logic. They use combinators to represent program transformations on learned programs, whereas we use combinators to represent components and generate families of programs that implement classes of ML pipelines. Moreover, our combinatory theory is typed, which is essential in our context for component- and search space specification and for ensuring the well-formedness of the generated models.

## 2. A Compositional Theory of Parametric Functions

In this section, we present an algebraic formulation of the **Para**-construction using string diagram constructors that will later serve as the basis for search space synthesis. The **Para**-construction defines a symmetric strict monoidal bicategory of parametric functions (Jia et al., 2025; Gavranović, 2024; Capucci et al., 2022; Gavranović et al., 2024) and provides a compositional theory for parametric functions. We will formalize the **Para**-construction as an algebraic theory by defining constructors for string diagrams, which represent morphisms in the **Para**-construction and, therefore, parametric functions.

To keep the topic accessible, we leave the mathematical details to Appendix A and present an algebraic theory of string diagrams for the **Para**-construction. This algebraic theory formalizes parametric functions as terms of string diagram constructors and term equalities, derived from the categorical definition of the **Para**-construction. The for-

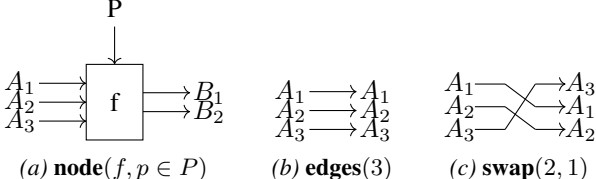

*(a)* **node**$(f, p \in P)$      *(b)* **edges**$(3)$      *(c)* **swap**$(2, 1)$

*Figure 1.* Node, edges, and swap constructors

malization follows the initial algebra approach of Gibbons (Gibbons, 1995) to construct an initial object in the category of enriched symmetric strict monoidal categories.

Following the categorical definitions, we explicitly distinguish between a product $P \bigotimes Q$ on parameters, a product $A \bigodot B$ on the domains and codomains of parametric functions, and a parameterization operation $\bullet$. They can, but do not necessarily need to be of the same product construction. A parametric function $f : A \bullet P \to B$ is a function from $A$ to $B$ parameterized over $P$. Note that every function can be viewed as a parametric function by choosing the parameter space to be the neutral element of $\bullet$ (usually the singleton set).

Consider affine transformations as an example. They are parametric functions $a : \mathbb{R}^n \bullet \mathbb{R}^{n \times m} \to \mathbb{R}^m$, implemented as matrix multiplication.

A parametric function $f : (\bigodot_{i \in [0,n]} A_i) \bullet P \to (\bigodot_{i \in [0,m]} B_i)$ corresponds to a node in a string diagram labeled with its identifier $f$ (see Figure 1a), with its domain as $n$ ingoing edges from the left, its codomain as $m$ outgoing edges to the right, and its parameters as an ingoing edge from above. If a function is parameterized by the neutral element of $\bullet$, we may omit the parameter edge. Note that $n$ and $m$ may be 0, corresponding to a node with no ingoing or outgoing edges. The string diagram constructor **node**$(f, p) : D_{(n,m)}$ is indexed by the function identifier $f$ and its choice of parameters $p \in P$. $D(n, m)$ is the type of string diagrams with $n$ ingoing and $m$ outgoing edges.

Now that we have defined the nodes of string diagrams, we define their edges. The string diagram constructor for $n$ parallel edges is **edges**$(n) : D_{(n,n)}$. Identity functions $\mathrm{id}_{(\bigodot_{i \in [1,n]} A_i)} : (\bigodot_{i \in [1,n]} A_i) \to (\bigodot_{i \in [1,n]} A_i)$ correspond to the edges of a string diagram; see Figure 1b. Note that identity functions do not allow $n$ to be zero, because **edges**$(0)$ encodes the empty string diagram, represented by the constructor $\epsilon : D_{(0,0)}$.

We are allowed to cross edges. However, we must do so explicitly by constructing a string diagram with **swap**$(n, m) : D_{(n+m,n+m)}$, crossing the top $n$ edges with the remaining $m$ edges. Note that $n$ and $m$ may be 0. Therefore, we derive the following equality of constructors:

$$\mathbf{swap}(n, 0) = \mathbf{swap}(0, n) = \mathbf{edges}(n)$$

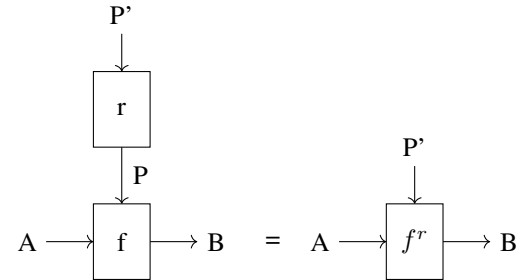

Figure 2. 2-Morphisms as reparameterizations

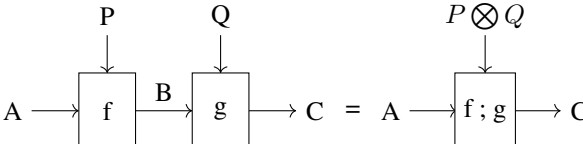

Figure 3. Sequential composition

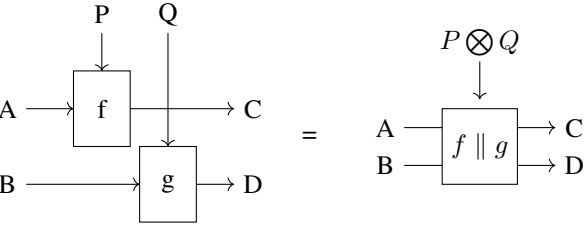

Figure 4. Parallel composition

$\mathbf{before}(\mathbf{beside}(f, g), \mathbf{beside}(h, k))$
$= \mathbf{beside}(\mathbf{before}(f, h), \mathbf{before}(g, k));$ (interchange law)
$\mathbf{before}(\mathbf{beside}(\mathbf{swap}(m, n), \mathbf{edges}(p)), \mathbf{beside}(\mathbf{edges}(n), \mathbf{swap}(m, p)))$
$= \mathbf{swap}(m, n + p);$ (swap law 1)
$\mathbf{before}(\mathbf{swap}(m, n), \mathbf{swap}(n, m))$
$= \mathbf{edges}(m + n);$ (swap law 2)
$\mathbf{before}(\mathbf{beside}(\mathbf{edges}(m), \mathbf{swap}(n, p)), \mathbf{beside}(\mathbf{swap}(m, p), \mathbf{edges}(n)))$
$= \mathbf{swap}(m + n, p);$ (swap law 3)
$\mathbf{repara}(r, \mathbf{node}(f, p)) = \mathbf{node}(f, r(p));$ (repara node)
$\cdots$

Figure 5. Term equations

This equality is already implicitly represented in our string diagram visualizations in Figure 1b and Figure 1c. It is possible to reparameterize a function. A reparameterization is the composition of a parameter function $r : P' \to P$ and a parametric function $f : A \bullet P \to B$, yielding a parametric function $f^r : A \bullet P' \to B$. This is depicted in Figure 2 as a string diagram, where the node $r$ is applied from above and constructed by $\mathbf{repara}(r) : D_{(n,m)} \to D_{(n,m)}$. This is our first constructor, which constructs a new string diagram from a given one. Reading the equality in Figure 2 from left to right shows how to *compose* multiple string diagrams into one, abstracting away unnecessary details. Reading the equality in Figure 2 from right to left shows how to *decompose* a single string diagram into its hidden components. String diagrams allow us to work at the right level of abstraction visually and, e.g., implicitly handle the associativity of constructors, whereas the corresponding terms must always contain all information and therefore enforce term equations to hold. Besides reparameterization, we have two more ways to compose given string diagrams into new ones. Figure 3 shows a string diagram that sequentially composes two parametric functions $f : A \bullet P \to B$ and $g : B \bullet Q \to C$ to construct a parametric function $f; g : A \bullet (P \otimes Q) \to C$. The constructor is $\mathbf{before}(p \otimes q) : D_{(n,k)} \to D_{(k,m)} \to D_{(n,m)}$, indexed by the composed parameters $p \otimes q \in P \otimes Q$ of the resulting parametric function. $\mathbf{edges}(n)$ is the neutral element for the sequential composition of string diagrams of type $D_{(n,m)}$ or $D_{(m,n)}$.

Parametric functions $f : A \bullet P \to C$ and $g : B \bullet Q \to D$ can also be composed in parallel to construct a parametric function $f || g : (A \odot B) \bullet (P \otimes Q) \to C \odot D$. The constructor is $\mathbf{beside}(p \otimes q) : D_{(n,m)} \to D_{(h,k)} \to D_{(n+h,m+k)}$, indexed over the composed parameters $p \otimes q \in P \otimes Q$ of the resulting parametric function. Parallel composition

of string diagrams is shown in Figure 4. The empty string diagram is the neutral element for parallel composition.

Now that we have introduced all the constructors for string diagrams, we need to define the term equations that encode the categorical laws for the **Para**-construction. For brevity, we leave the definition of all necessary term equations to Appendix A and focus in Figure 5 only on those that weren't already mentioned and aren't as obvious as, e.g., associativity laws for constructors or the remaining repara laws. For simplicity, we omit the indices of the constructors where they are obvious from the context. The swap laws are derived from the assumption that the **Para**-construction is symmetric. The interchange law follows from the fact that the **Para**-construction is a monoidal category. While the associativity and neutrality laws are implicitly encoded in all string diagram representations, Figure 6 and Figure 7 provide a visual interpretation of the laws from Figure 5. Note that the string diagrams defined here do not follow the usual definition of string diagrams for 2-categories. This is due to the **Para**-construction's definition of 2-morphisms as reparameterizations.

**Theorem 2.1.** *The algebraic theory defined by the presented constructor and term equations is a **Para**-construction.*

Proof in Appendix A.

## 3. Combinatory Logic Search Space Synthesis

The full definition of FCLP is given in (Dudenhefner et al., 2023). Here, we define only the concepts necessary for Section 4.

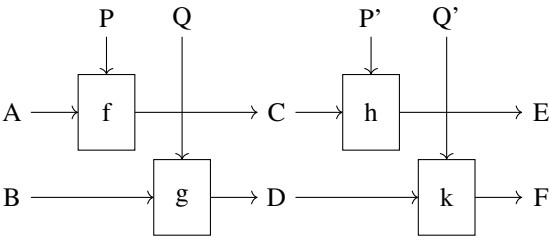

*Figure 6.* Symmetry law

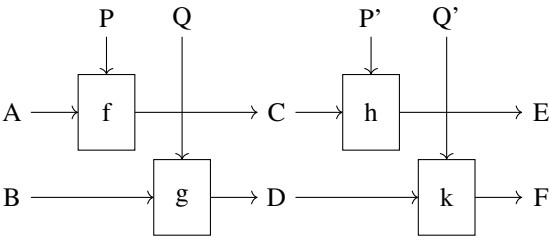

*Figure 7.* Interchange law

**Definition 3.1** (Intersection Types with Covariant Constructors and Literals)**.**

INTERSECTION TYPES

$$\mathbb{T} \ni \sigma, \tau, \rho ::= \omega \mid \sigma \to \tau \mid \sigma \cap \tau \mid C(\sigma) \mid l$$

where $C$ ranges over an enumerable set of *unary type constructors*, and $l$ ranges over an enumerable set of *literals*. For brevity, $C(\omega)$ is written $C$.

**Definition 3.2** (Parameterized Types)**.**

PARAMETERIZED TYPES

$$\varphi, \psi ::= \sigma \mid < y : t > \Rightarrow \varphi \mid \ll x : \sigma \gg \Rightarrow \varphi \mid P \Rightarrow \varphi$$

where $t$ ranges over collection identifiers and $P$ ranges over decidable predicates, possibly containing literal variables and term variables, ranged over by $x, y, z$. A literal variable $y$ is *bound* in $< y : t > \Rightarrow \varphi$. A term variable $x$ is *bound* in $\ll x : \sigma \gg \Rightarrow \varphi$.

For convenience, we extend the above definition of parameterized types with the following notational conventions:

$$< x : t > \Rightarrow < y : t > \Rightarrow \varphi$$
$$\leftrightarrow < x, y : t > \Rightarrow \varphi;$$
$$< y : t_1 > \Rightarrow < x : t_2 > \Rightarrow (x = f(y)) \Rightarrow \varphi$$
$$\leftrightarrow < y : t_1 > \Rightarrow < (x := f(y)) : t_2 > \Rightarrow \varphi;$$
$$< y : t_1 > \Rightarrow < x : t_2 > \Rightarrow < z : t_3 > \Rightarrow (z = f(x, y)) \Rightarrow \varphi$$
$$\leftrightarrow < y : t_1 > \Rightarrow < x : t_2 > \Rightarrow \varphi[z := f(x, y)];$$

**Definition 3.3** (Combinatory Terms and Arguments)**.**

| COMBINATORY TERMS | $\mathbb{C} \ni M, N ::= A \mid M\,T$ |
|---|---|
| COMBINATORY ARGUMENTS | $T ::= M \mid l$ |

where $A, B, C$ range over an enumerable set of *combinators*.

A type environment, also called a repository, assigns parametric intersection types to combinators. A literal environment assigns names to collections of literals. Combinatory Logic Synthesis (CLS) is the process of solving the relativized type inhabitation problem for FCLP. The Combinatory Synthesizer (CoSy) (Dudenhefner et al., 2026; Stahl, 2025) is a component-oriented software synthesis framework that implements a type-inhabitation algorithm for FCLP.

Given a repository, a literal environment, and a parameterized intersection type, CoSy computes a *parameterized tree grammar* (Stahl, 2025) that exactly describes the tree language of combinatory terms inhabiting the requested type. We call such grammars *solution spaces*. Combinatory terms may be interpreted in any domain or programming language, provided each combinator's interpretation adheres to the specification encoded in its parametric intersection type. For CLS, we therefore identify *components* as combinators and their *specifications* as parametric intersection types. CoSy provides methods for enumerating and interpreting terms from each computed solution space. However, this functionality alone is insufficient for applications that require optimizing a scoring function over combinatory terms. In these cases, the optimal term structure is usually unknown in advance. Our first contribution is a conservative extension of CoSy that equips synthesized solution spaces with sampling and search procedures. A *search space* is a solution space equipped with at least one sampling algorithm and search algorithms that are closed under the synthesized search space. The emptiness and finiteness problems for solution spaces are undecidable, yet semi-decidable (Dudenhefner et al., 2023), yielding an enumeration algorithm that is not guaranteed to terminate for every synthesized (possibly infinite) solution space. The same must be true for any sampling algorithm for search spaces. To address the feasibility of sampling, the following approach is proposed: For specifications that carefully use term predicates, the enumeration method can often decide the emptiness and finiteness problems. Sampling is performed under the user-guaranteed invariants that the search space is nonempty and large enough for the requested sample size. Users can test this invariant using the enumeration method. If enumeration does not produce a set of terms within a reasonable time, or if the search space is empty, both sampling and enumeration may fail to terminate. This is not considered a restriction, as generate-and-test remains feasible for small search spaces. Requiring users to verify the search space size first encourages consideration of generate-and-test methods when appropriate. The sampling method is a straightforward adaptation of a top-down sampling strategy from property-based testing (Hughes, 2007), with one modification: It will be shown later that a carefully defined specification can guarantee the finiteness of the search space. Accordingly, a Boolean flag is implemented to enable faster sampling

under the invariant that the search space is non-empty, finite, and contains a sufficient number of terms. Search methods closed under each synthesized search space are provided by adapting existing solutions for definite-clause grammars(Wong & Leung, 1997), and context-free grammars with constraints (Schrodi et al., 2023) to parameterized tree grammars. Two search methods are presented as prototypical examples. Evolutionary operations, specifically *crossover* and *mutation*, are defined on each synthesized search space by adapting algorithms from grammar-guided genetic programming on definite-clause grammars (Wong & Leung, 1997), that are closely related to parameterized tree grammars (Stahl, 2025). Because evolutionary algorithms require efficiently computable scoring functions, Bayesian optimization is proposed as a second search method for scoring functions that are computationally expensive. The approach from (Schrodi et al., 2023), which uses context-free grammars (CFGs) with constraints on the search space, can be adapted by replacing the CFGs with constraints and the evolutionary operators on them with parameterized tree grammars and the previously defined evolutionary operators. Given a kernel over combinatory terms, a Gaussian Process serves as a surrogate for the scoring function. The acquisition function is optimized using the evolutionary search method defined above. Recall that combinatory terms are language-agnostic and interpretable in any domain. This is particularly useful for graph kernels, as alternative graph interpretations allow the application of kernels without being restricted to the tree structure of combinatory terms. In Section 4, we will encode string diagrams that correspond to directed acyclic multigraphs (DAMG) as combinatory terms. Defining graph kernels based on the semantic DAMG structure and graph kernels on the tree structure of combinatory terms allows us to evaluate the practical relevance of our algebraic theory in Section 5.

Our extension of CoSy with evolutionary algorithms has already been adopted into the framework (Dudenhefner et al., 2026). Bayesian Optimization for CoSy is currently not part of the framework itself, but is part of the accompanying GitHub repository of elaborate examples (Laarmann, 2026a).

## 4. Modeling the Para-construction in FCLP

Our goal is to implement the algebraic theory from Section 2 as an FCLP repository to synthesize combinatory terms that represent string diagrams and to interpret them as parametric functions. The repository's combinators correspond to the constructors of string diagrams. We explain the steps to get from the repository in Figure 8, which is a plain translation of the constructors from Section 2 into typed combinators, to the repository in Figure 9, which models the algebraic theory, including equations.

$$\Gamma = \{$$
$$\mathbf{node}(f, p) : < n, m : \mathbb{N} > \Rightarrow$$
$$\mathrm{D}(\mathrm{IN}(n) \cap \mathrm{OUT}(m));$$
$$\mathbf{edges} : < n : \mathbb{N} > \Rightarrow$$
$$\mathrm{D}(\mathrm{IN}(n) \cap \mathrm{OUT}(n));$$
$$\mathbf{swap} : < n, m : \mathbb{N} > \Rightarrow$$
$$\mathrm{D}(\mathrm{IN}(n + m) \cap \mathrm{OUT}(n + m));$$
$$\mathbf{repara}(r) : < n, m : \mathbb{N} > \Rightarrow$$
$$\mathrm{D}(\mathrm{IN}(n) \cap \mathrm{OUT}(m)) \to \mathrm{D}(\mathrm{IN}(n) \cap \mathrm{OUT}(m));$$
$$\mathbf{before}(p \bigotimes q) : < n, m, k : \mathbb{N} > \Rightarrow \mathrm{D}(\mathrm{IN}(n) \cap \mathrm{OUT}(k))$$
$$\to \mathrm{D}(\mathrm{IN}(k) \cap \mathrm{OUT}(m)) \to \mathrm{D}(\mathrm{IN}(n) \cap \mathrm{OUT}(m));$$
$$\mathbf{beside}(p \bigotimes q) : < n, m, h, k : \mathbb{N} > \Rightarrow \mathrm{D}(\mathrm{IN}(n) \cap \mathrm{OUT}(m))$$
$$\to \mathrm{D}(\mathrm{IN}(h) \cap \mathrm{OUT}(k)) \to \mathrm{D}(\mathrm{IN}(n + h) \cap \mathrm{OUT}(m + k));$$
$$\epsilon : \mathrm{D}(\mathrm{IN}(0) \cap \mathrm{OUT}(0)); \}$$

*Figure 8.* String diagram constructors as an FCLP repository

The first point to note is that the signatures of indexed constructors already rely on the dependent type $D_{(n,m)}$ of string diagrams, indexed over their ingoing and outgoing edges, and are modeled as a literal abstraction over two literal variables $n, m \in \mathbb{N}$ in Figure 8. The second point is that we don't have a product construction for $(n, m)$ and therefore model it using intersection types by explicitly naming the projections IN and OUT, applying them as type constructors, and intersecting them. If we were to use the repository from Figure 8 for synthesis, we would compute search spaces with many string diagrams that are equivalent under the equations from the previous section, resulting in redundant interpretations of the combinatory terms. To overcome this, we will extend the parameterized intersection types to describe the quotient of combinatory terms under the equations of the algebraic theory. To do this, we utilize the approach from (Bessai et al., 2022), which restricts a tree grammar with a term rewriting system by forbidding its left-hand sides. Therefore, we direct the equations from left to right and describe how to synthesize only terms that match the right-hand sides.

**Lemma 4.1.** *The resulting term rewriting system is confluent.*

Proof in Appendix A.

To synthesize only terms matching the right-hand side of the associativity laws, we observe the list-like pattern of the terms on the right-hand side. Therefore, the proposed solution is a signature isomorphism that introduces two constructors for sequential and parallel composition, each operating in a list-like way. To enforce the right-hand side of the interchange law, we want to synthesize only terms that are sequential compositions of parallel compositions.

This can be done by stratifying the type of string diagrams $D(IN(n) \cap OUT(m))$ into three types $D_0(IN(n) \cap OUT(m))$ for constructors **node**, **edges**, **swap**, $D_1(IN(n) \cap OUT(m))$ for string diagrams that are parallel compositions of this constructors, and $D_2(IN(n) \cap OUT(m))$ for string diagrams that are sequential compositions of parallel compositions. The right-hand sides of the repara laws show that there is no normal form that contains a **repara**-constructor. Therefore, we must remove the **repara**-constructor from the repository. This makes sense, as combinators are indexed by their parameters, and reparameterizations are expressed by using the same combinator with a different argument. We must use the **edges**($n$) constructor rather than $n$ parallel compositions of **edges**(1) or any combination of different numbers of edges. Additionally, because edges are neutral elements under sequential composition, we need to track whether a parallel composition consists solely of edges. This is encoded as an intersection type that uses the types $ID, \overline{ID}, LAST(ID), LAST(\overline{ID})$ as labels, indicating whether a parallel composition is an identity ($ID, \overline{ID}$) and whether it is allowed to compose another **edges** constructor to ensure correct accumulation of **edges** under one constructor ($LAST(ID), LAST(\overline{ID})$). Since a **swap**($n, m$) constructor with $n \neq 0$ and $m \neq 0$ is not an identity, whereas **edges**($n$) is, we distinguish the two constructors and enforce a partition between them using predicates $P_{edges}$ and $P_{swap}$. The swap laws are challenging to encode using intersection types because they involve a deep pattern match on the constructor arguments. As a result, the left-hand sides of these laws are forbidden by term-predicates.

The goal of this paper is to define a general framework for synthesizing search spaces for parametric functions. The authors therefore argue that synthesizing a search space in which the empty string diagram is a valid solution is never intended. The proposed repository in Figure 9 will therefore explicitly exclude the construction of the empty string diagram and all corresponding laws. Some constructors in Figure 8 are still indexed; e.g., the **node** constructor is indexed by its function identifier and its parameters. We will introduce literal variables for these indices as well, and refine the intersection type to also encode this information. In Figure 9, all information the **node** constructor abstracts over is encoded in an extra literal variable $s$ and therefore is encoded in its parametric intersection type under the type constructor STRUCT. All constraints on $s$ are handled by the predicate $P_{node}$. We will distinguish three collections of literals for this encoding. The collection $\mathbb{L}_0$ will consist of triples, where the first projection identifies the concrete parametric function (the function identifier and parameters for **node**, special identifiers for **edges** and **swap**), and the second and third projections give the input and output numbers of edges in the constructed string diagram. The collection $\mathbb{L}_1$ consists of sequences of elements in $\mathbb{L}_0$ and serves as an encoding for parallel composition. The collection $\mathbb{L}_2$ will

consist of sequences of elements in $\mathbb{L}_1$ and will be used to encode sequential composition. This encoding scheme for all relevant string diagram information follows the composition structure and allows us to define parametric intersection types that precisely specify string diagrams in normal form. The consistency of this encoding, as well as the swap laws, is handled with predicates $P_{beside\_s}$, $P_{beside\_c}$, $P_{before\_s}$, and $P_{before\_c}$. With an additional predicate that determines whether two encodings are in the same equivalence class, one can even synthesize the normal form for encodings that are not in normal form and ensure that, for any encoding in the codomain, only the normal forms of that encoding will be requested in the domain.

**Theorem 4.2** (Uniqueness of encoding). *Any structure encoding of a normal form is isomorphic to its corresponding string diagram.*

Proof in Appendix A.
We can now provide a unique intersection type for each normal form of string diagrams. To precisely control variance in the synthesized search space, we introduce a special literal ¿ that encodes any valid structure at this position in the string diagram. We allow ¿ to appear in literals from $\mathbb{L}_0$ and $\mathbb{L}_1$, but not $\mathbb{L}_2$, forcing the structure of any sequential composition to be of a fixed length. We propagate each ¿ from the requested type through the compositional structure to the combinators that inhabit ¿ as well as to a specific literal, thereby obtaining a precise way to describe the variance of string diagrams in the search space. One may choose to allow ¿ for $\mathbb{L}_2$ as well, but this will yield an infinite search space. As mentioned in Section 3, we aim to model a finite search space to obtain a faster sampling algorithm.

**Lemma 4.3.** *The repository in Figure 9 specifies only finite search spaces if $\mathbb{L}_0$ and $\mathbb{L}_1$ are finite.*

Lemma 4.3 is a direct consequence of the previous argument. Because $\mathbb{L}_2$ may be infinite, we require each type to specify a finite sequence $s \in \mathbb{L}_2$. It doesn't matter how often ¿ occurs in $s$, the number of possible sequences obtained by unfolding ¿ in $s$ is finite, since the numbers of possible elements in $\mathbb{L}_0$ and $\mathbb{L}_1$ are finite. To reduce complexity and keep the definitions accessible, we restrict the encoding of variance in Figure 9 to elements of $\mathbb{L}_0$ and $\mathbb{L}_1$, as well as the numbers of ingoing and outgoing edges. Further variance can be encoded using the same principles. To conclude this section, we will discuss an example.
For readability, we write $(;)$ and $(||)$ as infix constructors for **before** and **beside** in the following definition of term $t$ that represents the string diagram in Figure 10. The term $t$ is the only inhabitant of the following intersection type $\sigma$ and can also be assigned the super type $\sigma'$ of $\sigma$.

$\Gamma = \{$
  **node** $: <f : \mathbb{F}> \Rightarrow <p : \mathbb{P}_f> \Rightarrow <n, m : \mathbb{N}> \Rightarrow$
    $<((s := ((f, p), n, m)) : \mathbb{L}_0)> \Rightarrow P_{\text{node}}(s) \Rightarrow$
    $D_0(\text{IN}(n) \cap \text{IN}(¿) \cap \text{OUT}(m) \cap \text{OUT}(¿) \cap$
    $\text{STRUCT}(s) \cap \text{STRUCT}(¿)) \cap \overline{\text{ID}};$
  **edges** $: <n : \mathbb{N}> \Rightarrow$
    $<((s := ((swap, n, 0), n, n)) : \mathbb{L}_0)>$
    $\Rightarrow P_{\text{edges}}(s) \Rightarrow$
    $D_0(\text{IN}(n) \cap \text{IN}(¿) \cap \text{OUT}(n) \cap \text{OUT}(¿) \cap$
    $\text{STRUCT}(s) \cap \text{STRUCT}(¿)) \cap \text{ID};$
  **swap** $: <n, m : \mathbb{N}> \Rightarrow$
    $<((s := ((swap, n, m), n + m, n + m)) : \mathbb{L}_0)> \Rightarrow$
    $P_{\text{swap}}(s) \Rightarrow$
    $D_0(\text{IN}(n + m) \cap \text{IN}(¿) \cap \text{OUT}(n + m) \cap \text{OUT}(¿) \cap$
    $\text{STRUCT}(s) \cap \text{STRUCT}(¿)) \cap \overline{\text{ID}};$
**beside_s** $: <n, m : \mathbb{N}> \Rightarrow <p : \mathbb{L}_0> \Rightarrow$
    $<(q := [p] : \mathbb{L}_1)> \Rightarrow P_{\text{beside\_s}}(n, m, p, q) \Rightarrow$
    $((D_0(\text{IN}(n) \cap \text{OUT}(m) \cap \text{STRUCT}(p)) \cap \overline{\text{ID}}) \rightarrow$
    $(D_1(\text{IN}(n) \cap \text{OUT}(m) \cap \text{STRUCT}(q) \cap$
    $\text{STRUCT}(¿)) \cap \overline{\text{ID}} \cap \text{LAST}(\overline{\text{ID}}))) \cap$
    $((D_0(\text{IN}(n) \cap \text{OUT}(m) \cap \text{STRUCT}(p)) \cap \text{ID}) \rightarrow$
    $(D_1(\text{IN}(n) \cap \text{IN}(¿) \cap \text{OUT}(m) \cap \text{OUT}(¿) \cap$
    $\text{STRUCT}(q) \cap \text{STRUCT}(¿)) \cap \text{ID}));$
**beside_c** $: <n, m, h, k : \mathbb{N}> \Rightarrow <p : \mathbb{L}_0> \Rightarrow$
    $<q : \mathbb{L}_1> \Rightarrow P_{\text{beside\_c}}(n, m, h, k, p, q) \Rightarrow$
    $((D_0(\text{IN}(n) \cap \text{OUT}(m) \cap \text{STRUCT}(p)) \cap \text{ID}) \rightarrow$
    $(D_1(\text{IN}(h) \cap \text{OUT}(k) \cap \text{STRUCT}(q))$
    $\cap \overline{\text{ID}} \cap \text{LAST}(\overline{\text{ID}})) \rightarrow$
    $(D_1(\text{IN}(n + h) \cap \text{IN}(¿) \cap \text{OUT}(m + k) \cap \text{OUT}(¿) \cap$
    $\text{STRUCT}([p] + q) \cap \text{STRUCT}(¿)) \cap \overline{\text{ID}} \cap \text{LAST}(\text{ID}))) \cap$
    $(((D_0(\text{IN}(n) \cap \text{OUT}(m) \cap \text{STRUCT}(p)) \cap \overline{\text{ID}})) \rightarrow$
    $(D_1(\text{IN}(h) \cap \text{OUT}(k) \cap \text{STRUCT}(q)) \cap \text{ID}) \rightarrow$
    $(D_1(\text{IN}(n + h) \cap \text{IN}(¿) \cap \text{OUT}(m + k) \cap \text{OUT}(¿) \cap$
    $\text{STRUCT}([p] + q) \cap \text{STRUCT}(¿)) \cap \overline{\text{ID}} \cap \text{LAST}(\overline{\text{ID}}))) \cap$
    $((D_0(\text{IN}(n) \cap \text{OUT}(m) \cap \text{STRUCT}(p)) \cap \overline{\text{ID}}) \rightarrow$
    $(D_1(\text{IN}(h) \cap \text{OUT}(k) \cap \text{STRUCT}(q)) \cap \overline{\text{ID}}) \rightarrow$
    $(D_1(\text{IN}(n + h) \cap \text{IN}(¿) \cap \text{OUT}(m + k) \cap \text{OUT}(¿) \cap$
    $\text{STRUCT}([p] + q) \cap \text{STRUCT}(¿)) \cap \overline{\text{ID}} \cap \text{LAST}(\overline{\text{ID}})));$
**before_s** $: <n, m : \mathbb{N}> \Rightarrow <p : \mathbb{L}_1> \Rightarrow <(q := [p] : \mathbb{L}_2)> \Rightarrow$
    $<<x : D_1(\text{IN}(n) \cap \text{OUT}(m) \cap \text{STRUCT}(p))>> \Rightarrow$
    $P_{\text{before\_s}}(n, m, p, q, x) \Rightarrow D_2(\text{IN}(n) \cap \text{IN}(¿) \cap$
    $\text{OUT}(m) \cap \text{OUT}(¿) \cap \text{STRUCT}(q));$
**before_c** $: <n, m, k : \mathbb{N}> \Rightarrow <a : \mathbb{L}_2> \Rightarrow$
    $<(b := normalize(a) : \mathbb{L}_2)> \Rightarrow$
    $<(p := head(b) : \mathbb{L}_1)> \Rightarrow <(q := tail(b) : \mathbb{L}_2)> \Rightarrow$
    $<<x : D_1(\text{IN}(n) \cap \text{OUT}(k) \cap \text{STRUCT}(p)) \cap \overline{\text{ID}}>> \Rightarrow$
    $<<y : D_2(\text{IN}(k) \cap \text{OUT}(m) \cap \text{STRUCT}(q))>> \Rightarrow$
    $P_{\text{before\_c}}(n, m, p, q, x, y) \Rightarrow$
    $D_2(\text{IN}(n) \cap \text{IN}(¿) \cap \text{OUT}(m) \cap \text{OUT}(¿) \cap \text{STRUCT}(a))$
    $\}$

*Figure 9.* The repository for our algebraic theory

$$[p_{(1,1)}] \bigotimes [p_{(2,1)} \bigotimes p_{(2,2)} \bigotimes p_{(2,3)}] \bigotimes [p_{(3,1)} \bigotimes p_{(3,2)}]$$

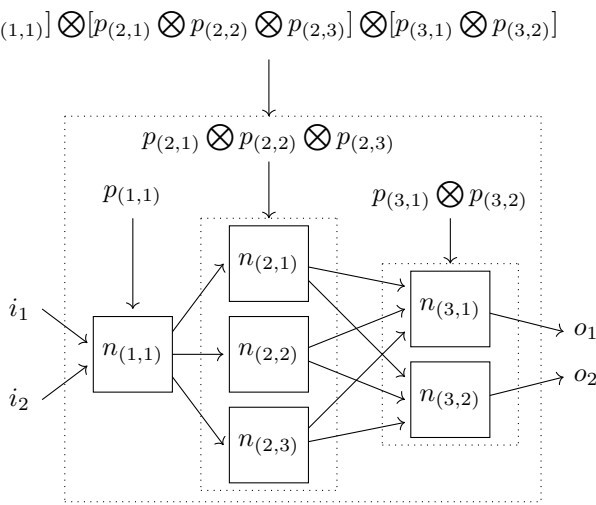

*Figure 10.* Neural network as string diagram

$$[p_{(1,1)}] \bigotimes [¿] \bigotimes [¿ \bigotimes p_{(3,2)}]$$

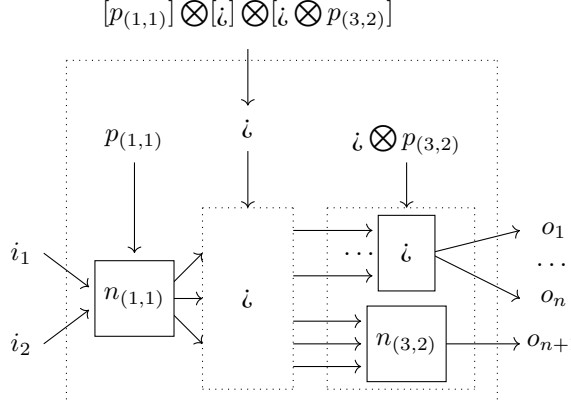

*Figure 11.* Structural variance in synthesized string diagrams

$t := \textbf{node}(n_{(1,1)}, p_{(1,1)})$ ;
    $(\textbf{node}(n_{(2,1)}, p_{(2,1)}) \,||\, \textbf{node}(n_{(2,2)}, p_{(2,2)}) \,||$
    $\textbf{node}(n_{(2,3)}, p_{(2,3)}))$ ;
    $(\textbf{edges}(1) \,||\, \textbf{swap}(1, 1) \,||\, \textbf{swap}(1, 1) \,||\, \textbf{edges}(1))$ ;
    $(\textbf{edges}(2) \,||\, \textbf{swap}(1, 1) \,||\, \textbf{edges}(2))$ ;
    $(\textbf{node}(n_{(3,1)}, p_{(3,1)}) \,||\, \textbf{node}(n_{(3,2)}, p_{(3,2)}))$

$\sigma := D_2(\text{IN}(2) \cap \text{OUT}(2) \cap \text{STRUCT}([[((n_{(1,1)}, p_{(1,1)}), 2, 3)],$
    $[((n_{(2,1)}, p_{(2,1)}), 1, 2), ((n_{(2,2)}, p_{(2,2)}), 1, 2),$
    $((n_{(2,3)}, p_{(2,3)}), 1, 2)],$
    $[((swap, 1, 0), 1, 1), ((swap, 1, 1), 2, 2),$
    $((swap, 1, 1), 2, 2), ((swap, 1, 0), 1, 1)],$
    $[((swap, 2, 0), 2, 2), ((swap, 1, 1), 2, 2),$
    $((swap, 2, 0), 2, 2)],$
    $[((n_{(3,1)}, p_{(3,1)}), 3, 1), ((n_{(3,2)}, p_{(3,2)}), 3, 1)]]))$

$$\sigma' := D_2(\text{IN}(2) \cap \text{OUT}(\textrm{¿}) \cap \text{STRUCT}([$$
$$[((n_{(1,1)}, p_{(1,1)}), 2, 3)],$$
$$\textrm{¿},$$
$$[((swap, 1, 0), 1, 1), ((swap, 1, 1), 2, 2),$$
$$((swap, 1, 1), 2, 2), ((swap, 1, 0), 1, 1)],$$
$$[((swap, 2, 0), 2, 2), ((swap, 1, 1), 2, 2),$$
$$((swap, 2, 0), 2, 2)],$$
$$[\,\textrm{¿}\,, ((n_{(3,2)}, p_{(3,2)}), 3, 1)]]]))$$

Therefore, the variance of string diagrams within a search space relies on the possibility of assigning more than one type to a term (see (Dudenhefner et al., 2023)). The intersection type $\sigma'$ describes a search space with multiple inhabitants and can also be depicted as a string diagram with holes, as seen in Figure 11.

# 5. Proof of Concept

We now evaluate the practical consequences of synthesizing search spaces using our algebraic theory for parametric functions. The experiments presented in this section are available through a GitHub repository (Laarmann, 2026b). The goal of this section is not to introduce a new optimization method, but to investigate how synthesized search spaces interact with standard optimization techniques such as kernel-based surrogate modeling and Bayesian optimization. To do so, we consider a simple one-dimensional process inspired by a physical bounce or reflection: an object moves linearly towards a boundary, changes direction at the boundary, and then moves linearly away from it. This gives rise to the piecewise-affine target function

$$f^\star(x) = \begin{cases} x + 10, & x \leq 0, \\ 10 - x, & x > 0. \end{cases}$$

Equivalently, on the considered domain, this corresponds to the function $10 - |x|$, which exhibits a sharp change in direction at $x = 0$. The relevance of this example lies not in the difficulty of the regression problem but in its compositional structure: the target mechanism consists of a splitting decision followed by two affine branches. For evaluation, we generate training samples by evaluating $f^\star$ on $N_{\text{train}} = 1000$ points from the interval $[-10, 10]$, adding small Gaussian noise $\varepsilon \sim \mathcal{N}(0, 0.05)$ to account for moderate observation noise. To evaluate whether a learned model captures the underlying mechanism rather than merely interpolating the observed samples, we evaluate the model on $N_{\text{test}} = 1000$ points from the larger interval $[-15, 15]$. The test interval, therefore, contains out-of-distribution regions not observed during training. To synthesize search spaces from the repository in Figure 9, we first need to define the components from which we will construct parametric functions. As components, we want to

use a selection of PyTorch modules, which are embedded in the repository by providing the literal collections $\mathbb{F}$ and $\mathbb{P}_f$ to define the literal collection $\mathbb{L}_0$, uniquely identifying each valid node. Therefore, the identifiers in $\mathbb{F} = \{\text{Linear, Sigmoid, ReLU, Tanh, Sum, Product, Copy}\}$ correspond to PyTorch module names and tensor operations. $\mathbb{P}_f$ then contains valid hyperparameters for each module identifier $f \in \mathbb{F}$ that comply with the PyTorch module parameters. Consequently, any synthesized term can be straightforwardly interpreted as a PyTorch module by pattern matching on the fifth literal argument from $\mathbb{L}_0$ of the node constructor. The other constructors from the repository are implemented as the corresponding gating, branching, and composition operators from PyTorch. For our experiment, we set the literal collection $\mathbb{N} = \{1, 2, 3, 4, 5\}$ and consider various search spaces, all uniquely defined by an intersection type $\tau$ to control their variance and restrict their size.

$$\tau_3 := D_2(\text{IN}(1) \cap \text{OUT}(1) \cap \text{STRUCT}([\textrm{¿}, \textrm{¿}, \textrm{¿}]))$$
$$\tau_{3\text{-refined}} := D_2(\text{IN}(1) \cap \text{OUT}(1) \cap \text{STRUCT}([[\textrm{¿}], \textrm{¿}, [\textrm{¿}]]))$$
$$\cdots$$
$$\tau_{5\text{-refined-2}} := D_2(\text{IN}(1) \cap \text{OUT}(1) \cap \text{STRUCT}([[\textrm{¿}], [\textrm{¿}, \textrm{¿}, \textrm{¿}], \textrm{¿}, [\textrm{¿}, \textrm{¿}], [\textrm{¿}]]))$$

Each term is interpreted as a PyTorch module implementing the parametric function, which is trained on the training data for 2000 epochs, and its loss is computed on the test data. Therefore, our objective function $f_{\text{obj}}$ is the interpretation of the term as the test loss of the parametric function, which corresponds to the term, compared to $f^\star$. We use a Gaussian process regressor as the surrogate model. The kernel determines the surrogate's capacity to approximate the objective function. We analyze how kernel choice influences model fidelity. For each synthesized search space, we sample 100 terms and quantify kernel–objective alignment by the Spearman rank correlation between pairwise kernel similarities and negative objective differences,

$$\rho(k(x_i, x_j), \ -|f_{\text{obj}}(x_i) - f_{\text{obj}}(x_j)|),$$

so that higher values indicate stronger agreement between kernel similarity and objective proximity. To reduce noise, the kernel–objective alignment is computed using the mean objective value across multiple training seeds. We compare three kernel constructions: a subtree kernel applied to the combinatory terms, a Weisfeiler-Lehman kernel applied to the same terms, and a Weisfeiler-Lehman kernel applied to the DAMG obtained by interpreting a term as a string diagram. The latter exploits the categorical semantics of the synthesized models by operating on the structural representation induced by the Para-construction. Across all considered search spaces, kernels operating on DAMGs consistently exhibit stronger alignment with the objective than kernels applied directly to the syntactic tree representation of the combinatory terms, as can be seen in Figure 12.

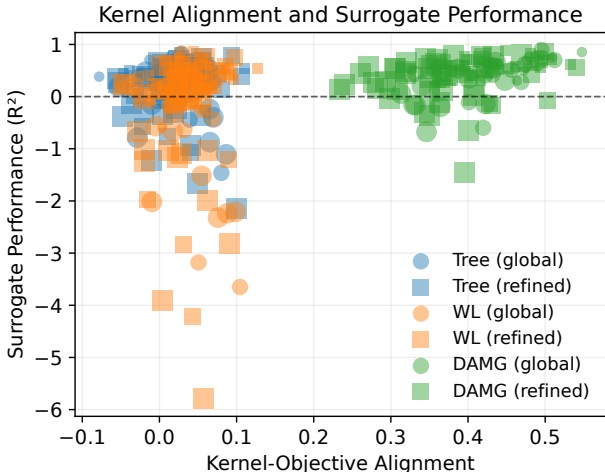

*Figure 12.* Kernel-objective alignment vs. surrogate learning performance

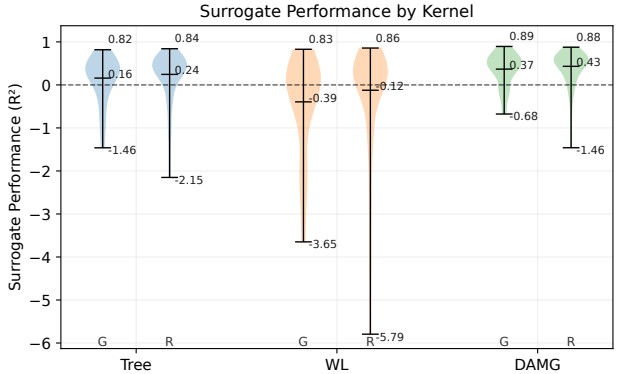

*Figure 13.* Surrogate learning performance across kernels. G: maximal variance; R: restricted variance

| Kernel | Space | NMSE ↓ | | $R^2$ ↑ | |
|---|---|---|---|---|---|
| | | Mean | Max | Mean | Max |
| Tree | Global | 0.837 | 2.752 | 0.158 | 0.817 |
| | Refined | 0.946 | 4.559 | 0.244 | 0.841 |
| WL | Global | 1.035 | 3.032 | -0.394 | 0.827 |
| | Refined | 1.046 | 4.845 | -0.124 | 0.856 |
| DAMG | Global | **0.781** | 2.921 | 0.367 | 0.893 |
| | Refined | 0.874 | 4.664 | **0.431** | 0.876 |

*Table 1.* Mean and max NMSE ($\downarrow$) and $R^2(\uparrow)$ for Tree, WL, and DAMG kernels. Best mean values in bold; best maxima underlined

the same evaluation budget. In all configurations, Bayesian optimization identifies better candidates than random search within the given budget. At the same time, the optimization performance varies across the considered specifications: certain refinements improve optimization performance, while others restrict the search space in ways that make the optimization problem more difficult. The best candidate found across all experiments for $f_{\text{obj}}$ exhibits an unconventional model structure. We discuss it in Appendix C. Overall, these experiments demonstrate that synthesized search spaces can be explored effectively using standard optimization techniques. Moreover, they indicate that the categorical structure underlying the string diagram representation provides a description well-suited to kernel-based surrogate modeling. These results support the view that categorical structure provides a useful inductive bias for defining and exploring search spaces of parametric functions.

## 6. Conclusion

This paper introduces a framework for synthesizing search spaces of parametric functions from typed component specifications. We provide an FCLP repository encoding the algebraic theory of parametric functions induced by the Para-construction and extend the CoSy framework so that synthesized solution spaces can serve as search spaces equipped with sampling and optimization procedures. The framework enables the automatic construction of search spaces from component specifications rather than requiring manually designed architecture grammars. Our proof-of-concept implementation demonstrates that the synthesized spaces can be explored using standard optimization techniques such as kernel-based surrogate modeling and Bayesian optimization. Our experiments indicate that the string diagram representation induced by the categorical model provides structural information that is beneficial for kernel-based modeling of the objective function. From a theoretical perspective, an important topic for future work is categorical semantics for FCLP allowing formal reasoning about soundness and completeness of models of parametric functions.

This suggests that the structural information captured by the categorical string diagram representation is more closely related to the functional behavior of the synthesized models than the purely syntactic tree structure of the combinatory terms. We next evaluate whether the objective function can be effectively approximated by surrogate models defined over the synthesized search spaces. For each kernel, we train a Gaussian process regressor on sampled candidates and evaluate predictive performance using normalized mean squared error and the $R^2$ score. Gaussian processes using the DAMG kernel consistently achieve lower prediction error than the corresponding tree-based kernels, see Table 1. Furthermore, the refined search spaces tend to exhibit improved predictive accuracy compared to their less-restricted counterparts, as shown in Figure 13. This indicates that restricting structural variance can improve the learnability of the objective over the synthesized search space. In Appendix B, we evaluate Bayesian optimization on the synthesized search spaces and compare it to random search under

## Impact Statement

This paper presents work whose goal is to advance the field of Machine Learning. There are many potential societal consequences of our work, none of which we feel must be specifically highlighted here.

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

# A. Category-Theoretic Background and Proofs

This appendix provides the categorical background and formal proofs underlying the algebraic theory of parametric functions used in the main text.

We assume a basic knowledge of category theory for this section, or recommend reading (Mac Lane, 1998).

## A.1. The Para-construction

We restrict attention to strict monoidal categories, which allow us to express natural isomorphisms as equations.

**Definition A.1** ($\mathcal{M}$-Actegory (compare (Gavranović, 2024))). Let $(\mathcal{M}, \bigotimes, I)$ be a strict monoidal category. A strict $(\mathcal{M}, \bigotimes, I)$-actegory is a category $\mathcal{C}$ together with a functor $\bullet : \mathcal{M} \times \mathcal{C} \to \mathcal{C}$, and two natural isomorphisms called *the unitor* $\eta_C : I \bullet C = C$ and *the multiplicator* $\mu_{C,M,N} : C \bullet (M \bigotimes N) = (C \bullet M) \bullet N$ whose components are objects $C : \mathcal{C}$ and $M, N : \mathcal{M}$, satisfying the following coherence laws:

- Pentagon law: For all $C : \mathcal{C}$ and $M, N, P : \mathcal{M}$, $C \bullet (M \bigotimes (N \bigotimes P)) = ((C \bullet M) \bullet N) \bullet P$.

- Unit laws: For all $C : \mathcal{C}$ and $M : \mathcal{M}$, $C \bullet (M \bigotimes I) = C \bullet M$ and $C \bullet (I \bigotimes M) = C \bullet M$

**Definition A.2** (**Para**-construction (compare (Gavranović, 2024))). Let $(\mathcal{M}, \bigotimes, I)$ be a strict monoidal category and $(\mathcal{C}, \bullet)$ be a strict $(\mathcal{M}, \bigotimes, I)$-actegory. We define the bicategory **Para**$_\bullet(\mathcal{C})$ of parametric morphisms as follows:

- The objects are those of $\mathcal{C}$;

- The morphisms of type $A \to B$ are pairs $(P, f)$ for parameter spaces $P : \mathcal{M}$, morphisms $f : A \bullet P \to B$, and objects $A, B : \mathcal{C}$;

- The 2-morphisms of type $(P, f) \Rightarrow (P', f')$ are morphisms $r : P' \to P$ in $\mathcal{M}$ such that the following diagram

commutes:

$$
\begin{array}{ccc}
A \bullet P' & \xrightarrow{\ f'\ } & B \\
{\scriptstyle A \bullet r}\Big\downarrow & \nearrow{\scriptstyle f} & \\
A \bullet P & &
\end{array}
$$

- The identities are morphisms $(I, \eta_A)$ for every object $A : \mathcal{C}$;

- The sequential composition of morphisms $(P, f)$ and $(Q, g)$ yields a morphism $(P \bigotimes Q, f; g)$ where $(f; g) : A \bullet (P \bigotimes Q) \to C$ in $\mathcal{C}$.

- Horizontal and vertical composition of 2-morphisms are given by the monoidal structure of $\mathcal{M}$.

**Definition A.3** (Symmetric Strict Monoidal Actegory (compare (Gavranović, 2024))). Let $(\mathcal{M}, \bigotimes, I)$ be a symmetric strict monoidal category. A $\mathcal{M}$-actegory $(\mathcal{C}, \bullet)$ is called a symmetric strict monoidal $\mathcal{C}$-actegory if the underlying category $\mathcal{C}$ has a symmetric monoidal structure $(\bigodot, J)$ and the parameterization operation $\bullet$ respects that structure. This means that

- The underlying functor $\mathcal{C} \times \mathcal{M} \to \mathcal{C}$ is strong monoidal;

- The underlying natural isomorphisms $\eta$ and $\mu$ are monoidal;

**Definition A.4** (Symmetric Strict Monoidal Para-construction (compare (Gavranović, 2024))). Let $(\mathcal{C}, \bullet)$ be a symmetric strict monoidal $\mathcal{M}$-actegory with the underlying symmetric monoidal structure $(\bigodot, J)$. Then **Para**$_\bullet(\mathcal{C})$ becomes a symmetric strict monoidal bicategory with the following constructions additionally to Definition A.2:

- The monoidal product of any two objects $A, B : \mathcal{C}$ is given by $A \bigodot B$ of $(\mathcal{C}, \bigodot, J)$;

- The monoidal unit is the monoidal unit of $(\mathcal{C}, \bigodot, J)$;

- The monoidal product of two morphisms $(P, f : A \bullet P \to B)$ and $(Q, g : C \bullet Q \to D)$ is given by $(P \bigotimes Q, f \bigodot g : (A \bigodot C) \bullet (P \bigotimes Q) \to B \bigotimes D)$.

- The braiding operation $\beta_{A,B} : A \odot B \to B \odot A$;

- The symmetry law $\beta_{A,B} ; \beta_{B,A} = (I, \eta_{A \odot B})$ must hold;

- The interchange law
  $(P, f) \odot (P', h); ((Q, g) \odot (Q', i)) = ((P, f); (Q, g)) \odot ((P', h); (Q', i))$ must hold;

## A.2. Algebraic Theory

Let us summarize the results of Section 2 as follows:

**Definition A.5** (Signature)**.**

$$
\Sigma = \{
$$
$$
\mathbf{node}(n, m, f, p) : \ \mathrm{D}_{(n,m)}
$$
$$
\mathbf{edges}(n) : \ \mathrm{D}_{(n,n)}
$$
$$
\mathbf{swap}(n, m) : \ \mathrm{D}_{(n+m,n+m)}
$$
$$
\mathbf{repara}(n, m, r) : \ \mathrm{D}_{(n,m)}
$$
$$
\mathbf{before}(n, k, m, p \bigotimes q) : \ \mathrm{D}_{(n,k)} \to \mathrm{D}_{(k,m)} \to \mathrm{D}_{(n,m)}
$$
$$
\mathbf{beside}(n, m, h, k, p \bigotimes q) : \ \mathrm{D}_{(n,m)} \to \mathrm{D}_{(h,k)} \to \mathrm{D}_{(n+h,m+k)}
$$
$$
\epsilon : \mathrm{D}_{(0,0)}
$$
$$
\}
$$

We define term variables $x_{(n,m)} : D_{(n,m)} \in V$ indexed over the incoming and outgoing edges to enforce type correctness for the indexed sorts $D_{(n,m)}$. We omit the indices, if they are clear from the context.

**Definition A.6** (Term Equations)**.**

$$\mathbf{repara}(r, \mathbf{node}(n, m, f, p)) = \mathbf{node}(n, m, f, r(p)) \tag{repara node}$$
$$\mathbf{repara}(r, \mathbf{edges}(n)) = \mathbf{edges}(n) \tag{repara edges}$$
$$\mathbf{repara}(r, \mathbf{swap}(n, m)) = \mathbf{swap}(n, m) \tag{repara swap}$$
$$\mathbf{repara}(r, \mathbf{before}(n, k, m, (p, q))(x, y)) = \mathbf{before}(n, k, m, r((p, q)))(x, y) \tag{repara before}$$
$$\mathbf{repara}(r, \mathbf{beside}(n, m, h, k, (p, q))(x, y)) = \mathbf{beside}(n, m, h, k, r((p, q)))(x, y) \tag{repara beside}$$
$$\mathbf{repara}(r, \epsilon) = \epsilon \tag{repara empty}$$
$$\mathbf{before}(\mathbf{beside}(v, w), \mathbf{beside}(x, y)) = \mathbf{beside}(\mathbf{before}(v, x), \mathbf{before}(w, y)) \tag{interchange law}$$
$$\mathbf{beside}(\mathbf{before}(x, \epsilon), \mathbf{before}(\epsilon, y)) = \mathbf{before}(\mathbf{beside}(x, \epsilon), \mathbf{beside}(\epsilon, y)) \tag{dislocation law}$$
$$\mathbf{before}(\mathbf{beside}(\mathbf{swap}(m, n), \mathbf{edges}(p)), \mathbf{beside}(\mathbf{edges}(n), \mathbf{swap}(m, p))) = \mathbf{swap}(m, n + p) \tag{swap law 1}$$
$$\mathbf{before}(\mathbf{swap}(m, n), \mathbf{swap}(n, m)) = \mathbf{edges}(m + n) \tag{swap law 2}$$
$$\mathbf{before}(\mathbf{beside}(\mathbf{edges}(m), \mathbf{swap}(n, p)), \mathbf{beside}(\mathbf{swap}(m, p), \mathbf{edges}(n))) = \mathbf{swap}(m + n, p) \tag{swap law 3}$$
$$\mathbf{before}(\mathbf{swap}(m, n), \mathbf{before}(\mathbf{beside}(x_{(n,p)}, y_{(m,q)}), \mathbf{swap}(p, q))) = \mathbf{beside}(x_{(n,p)}, y_{(m,q)}) \tag{swap law 4}$$
$$\mathbf{beside}(\mathbf{beside}(x, y), z) = \mathbf{beside}(x, \mathbf{beside}(y, z)) \tag{associativity of beside}$$
$$\mathbf{before}(\mathbf{before}(x, y), z) = \mathbf{before}(x, \mathbf{before}(y, z)) \tag{associativity of before}$$
$$\mathbf{before}(\mathbf{edges}(n), x_{(n,m)}) = x_{(n,m)} \tag{left unit before}$$
$$\mathbf{before}(x_{(n,m)}, \mathbf{edges}(m)) = x_{(n,m)} \tag{right unit before}$$
$$\mathbf{beside}(\epsilon, x) = x \tag{left unit beside}$$
$$\mathbf{beside}(x, \epsilon) = x \tag{right unit beside}$$
$$\mathbf{beside}(\mathbf{edges}(n), \mathbf{edges}(m)) = \mathbf{edges}(n + m) \tag{edges beside}$$

**Definition A.7** (Algebraic Theory). The signature $\Sigma$ from Definition A.5 together with the term equations in Definition A.6 defines an algebraic theory.

**Theorem A.8.** *The algebraic theory is a symmetric strict monoidal **Para**-construction.*

*Proof.* Note that string diagrams are directed acyclic multigraphs (DAMGs). The proof is analogous to Gibbons proof (Gibbons, 1995) that his algebraic theory for DAMGs is an enriched symmetric strict monoidal category.
We align the constructions and laws from Definition A.2 and Definition A.4 with the term constructors of $\Sigma$ in Definition A.5 and the term equations in Definition A.6. We assume a given symmetric strict monoidal $\mathcal{M}$-actegory $(\mathcal{C}, \bullet)$, with morphisms indexed by identifiers $f, g, \ldots$ and choose them together with $\mathcal{M}$ as the indices for the constructors.

- Objects $A$ are isomorphic to identities $(I, \eta_A)$ and correspond to the **edges**-constructor;

- Morphisms $(P, f)$ correspond to the **node**$(f, p)$-constructor, that is indexed over the function identifier $f$ and it's parameters $p \in P$;

- The 2-morphisms $r : P' \to P$ correspond to the **repara**$(r)$-constructor;

- The sequential composition corresponds to the **before**-constructor;

- The horizontal and vertical composition of 2-morphisms corresponds to the repara laws;

- The monoidal product corresponds to the **beside**-constructor;

- The monoidal unit corresponds to the $\epsilon$-constructor;

- The braiding operation corresponds to the **swap**-constructor;

- Para being a bicategory corresponds to the associativity and unit laws;

- The symmetry law corresponds to the swap laws;

- The interchange law corresponds to the interchange law.

- The dislocation law is a direct consequence of the interchange law.

$\square$

## A.3. Derived properties

The complete term rewriting system for Section 4 is the following:

**Definition A.9** (Term Rewriting System)**.** The following term rewriting system is obtained by directing the equation from Definition A.6.

$$\text{TRS} := \{$$

$$\textbf{repara}(r, \textbf{node}(n, m, f, p)) \rightarrow \textbf{node}(n, m, f, r(p));$$

$$\textbf{repara}(r, \textbf{edges}(n)) \rightarrow \textbf{edges}(n);$$

$$\textbf{repara}(r, \textbf{swap}(n, m)) \rightarrow \textbf{swap}(n, m);$$

$$\textbf{repara}(r, \textbf{before}(n, k, m, (p, q))(x, y)) \rightarrow \textbf{before}(n, k, m, r((p, q)))(x, y);$$

$$\textbf{repara}(r, \textbf{beside}(n, m, h, k, (p, q))(x, y)) \rightarrow \textbf{beside}(n, m, h, k, r((p, q)))(x, y);$$

$$\textbf{repara}(r, \epsilon) \rightarrow \epsilon;$$

$$\textbf{before}(\textbf{beside}(v, w), \textbf{beside}(x, y)) \rightarrow \textbf{beside}(\textbf{before}(v, x), \textbf{before}(w, y));$$

$$\textbf{beside}(\textbf{before}(x, \epsilon), \textbf{before}(\epsilon, y)) \rightarrow \textbf{before}(\textbf{beside}(x, \epsilon), \textbf{beside}(\epsilon, y));$$

$$\textbf{before}(\textbf{beside}(\textbf{swap}(m, n), \textbf{edges}(p)), \textbf{beside}(\textbf{edges}(n), \textbf{swap}(m, p))) \rightarrow \textbf{swap}(m, n + p);$$

$$\textbf{before}(\textbf{swap}(m, n), \textbf{swap}(n, m)) \rightarrow \textbf{edges}(m + n);$$

$$\textbf{before}(\textbf{beside}(\textbf{edges}(m), \textbf{swap}(n, p)), \textbf{beside}(\textbf{swap}(m, p), \textbf{edges}(n))) \rightarrow \textbf{swap}(m + n, p);$$

$$\textbf{before}(\textbf{swap}(m, n), \textbf{before}(\textbf{beside}(x_{(n,p)}, y_{(m,q)}), \textbf{swap}(p, q))) \rightarrow \textbf{beside}(x_{(n,p)}, y_{(m,q)});$$

$$\textbf{beside}(\textbf{beside}(x, y), z) \rightarrow \textbf{beside}(x, \textbf{beside}(y, z));$$

$$\textbf{before}(\textbf{before}(x, y), z) \rightarrow \textbf{before}(x, \textbf{before}(y, z));$$

$$\textbf{before}(\textbf{edges}(n), x_{(n,m)}) \rightarrow x_{(n,m)};$$

$$\textbf{before}(x_{(n,m)}, \textbf{edges}(m)) \rightarrow x_{(n,m)};$$

$$\textbf{beside}(\epsilon, x) \rightarrow x;$$

$$\textbf{beside}(x, \epsilon) \rightarrow x;$$

$$\textbf{beside}(\textbf{edges}(n), \textbf{edges}(m)) \rightarrow \textbf{edges}(n + m);$$

$$\}$$

**Lemma A.10.** *TRS is confluent.*

*Proof.* The term rewriting system TRS is linear because each variable appears at most once on both the left-hand and right-hand sides. Only the unit rules overlap with the laws of interchange, dislocation, and associativity. Therefore, Huet's strong closedness criterion (Huet, 1980) applies, and the TRS is confluent. □

**Theorem A.11** (Uniqueness of encoding)**.** *The encoding of the structure of the parameter space within literals from the collection $\mathbb{L}_2$ in Section 4 is isomorphic to the inhabitants.*

Recall the literal collections from Section 4:

- $\mathbb{L}_0$ contains literals $((f, p), n, m)$ for nodes and $((swap, n, m))$ for swaps and edges;

- $\mathbb{L}_1$ are sequences of literals from $\mathbb{L}_0 \cup \{?'\}$;

- $\mathbb{L}_2$ are sequences of literals from $\mathbb{L}_1 \cup \{?'\}$;

**Proof sketch:** The proof involves inductively defining a function from terms over $\Gamma$ (see Section 4 for the definition) to literals of $\mathbb{L}_2$.

A function that maps the literals of $\mathbb{L}_2$ to terms over $\Gamma$ can be defined by induction over $\mathbb{L}_2$.

Note that term constructors take literal arguments from $\mathbb{L}_0, \mathbb{L}_1, \mathbb{L}_2$ in a way that makes defining these functions straightforward. At last, one must prove that these functions are inverse to each other. This can be proven by structural induction, again using the literal arguments of the constructors.

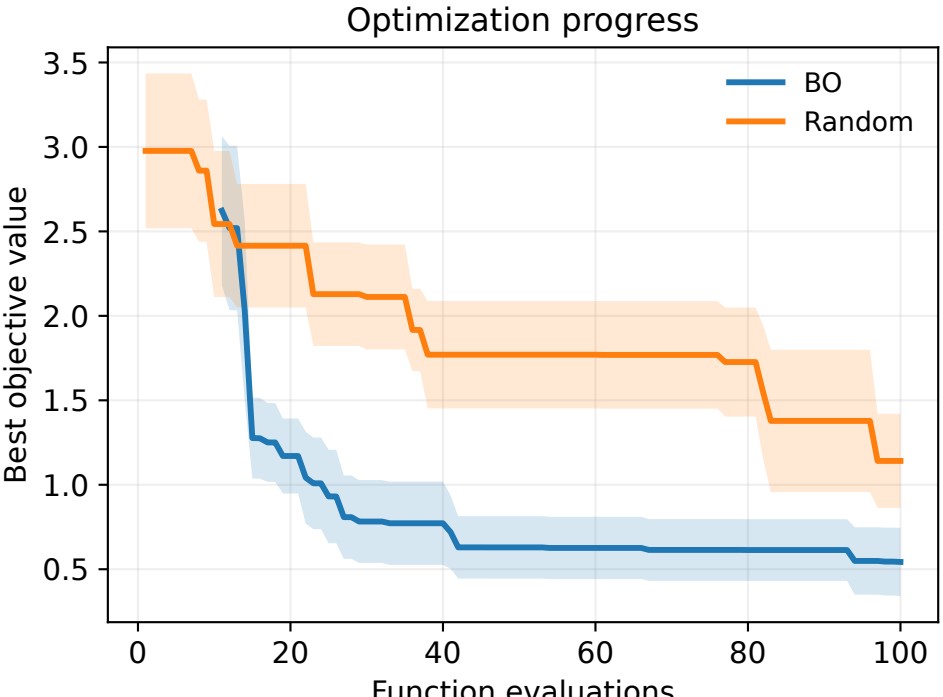

*Figure 14.* Optimization progress of Bayesian optimization and random search across multiple synthesized search spaces

## B. Additional Optimization Experiments

In addition to the surrogate modeling analysis in Section 5, we evaluate the effectiveness of Bayesian optimization (BO) for exploring the synthesized search spaces. In this experiment, BO is compared against random search under the same evaluation budget. For each considered search space specification, candidate models are sampled and evaluated using the same training and testing procedure described in Section 5. Bayesian optimization uses the Gaussian process regressor together with an optimizable DAMG kernel defined over the synthesized model structures. Random search samples candidates uniformly from the same search space.

Figure 14 shows the optimization progress measured by the best observed objective value as a function of the number of function evaluations. Each curve aggregates optimization runs across multiple synthesized search spaces. Bayesian optimization is initialized with a sample of 10 terms; therefore its curve starts at x = 10, whereas random search starts at x = 0. Across all configurations, Bayesian optimization consistently identifies better candidates within the same evaluation budget compared to random search. This confirms that the synthesized search spaces can be effectively explored using standard surrogate-based optimization methods such as Bayesian optimization.

$$[((\text{Copy}, 4), 1, 4)] \bigotimes [((\text{ReLU}, ()), 2, 2) \bigotimes ((\text{Linear}, (2, 1, \text{True})), 2, 1)] \bigotimes [((\text{Linear}, (4, 1, \text{True})), 4, 1)]$$

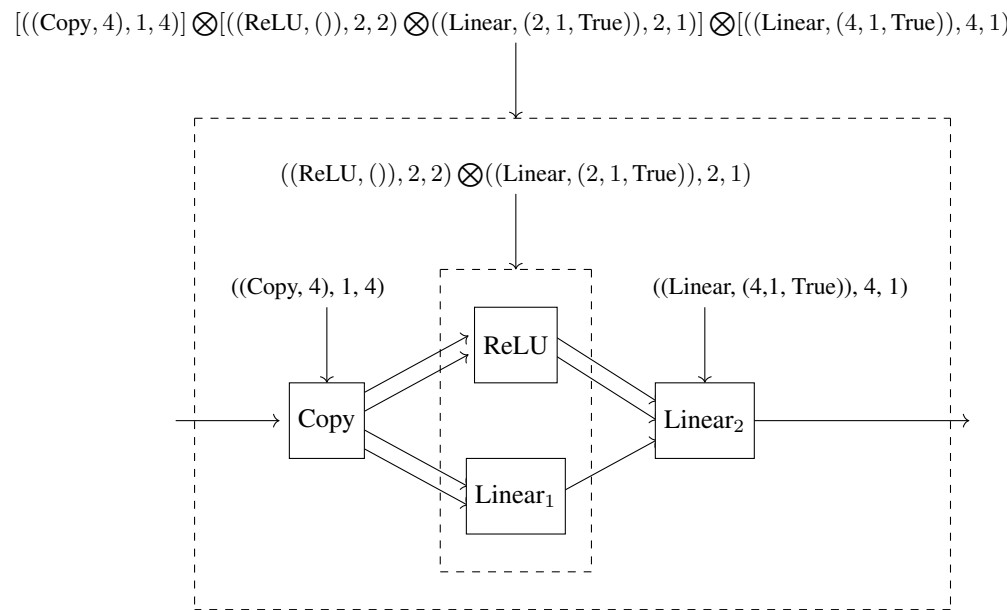

*Figure 15.* Best candidate discovered during experiments

## C. Best Candidate Discovered by Bayesian Optimization

Figure 15 visualizes the best candidate discovered during the Bayesian optimization experiments described in Section 5. Since the structure of this candidate is known, it can be reproduced by inhabiting the following intersection type:

$$\tau := D_2(\text{IN}(1) \cap \text{OUT}(1) \cap$$
$$\text{STRUCT}([((\text{Copy}, 4), 1, 4)] \bigotimes [((\text{ReLU}, ()), 2, 2) \bigotimes ((\text{Linear}, (2, 1, \text{True})), 2, 1)] \bigotimes [((\text{Linear}, (4, 1, \text{True})), 4, 1)]))$$

The model is represented as a string diagram corresponding to the synthesized combinatory term. The resulting architecture is surprisingly compact and consists of a small number of components. When interpreted as a PyTorch module, the model implements the function

$$f^\bullet(x) = (\max(0, x), \max(0, x), (x, x)A_1^T + b_1)A_2^T + b_2$$

where $A_1$ and $A_2$ are the weight matrices of the linear layers and $b_1, b_2$ are their corresponding biases. The candidate is plotted against the test data in Figure 16.

Notably, the architecture applies the ReLU activation directly to the inputs rather than to the outputs of a linear transformation. While this design may appear unusual from a conventional neural network perspective, it provides an effective approximation of the target function $f^\star$, which contains a piecewise-linear structure with a change in slope at $x = 0$. To make this clearer, we assign the parameters of the trained model to $f^\bullet$, which leads to the function

$$f^\bullet(x) = (-1.1067 \cdot \max(0, x)) + (-0.8939 \cdot \max(0, x)) + (0.5465x + 0.4545x + 8.5420) + 1.4591$$

or equivalently written with a case distinction similar to $f^\star$

$$f^\bullet(x) = \begin{cases} (0.5465x + 0.4545x + 8.5420) + 1.4591, & x \leq 0, \\ (-1.1067 \cdot \max(0, x)) + (-0.8939 \cdot \max(0, x)) + (0.5465x + 0.4545x + 8.5420) + 1.4591, & x > 0 \end{cases}$$

which simplifies to

$$f^\bullet(x) = \begin{cases} 1.001x + 10.0011, & x \leq 0, \\ -0.9996x + 10.0011, & x > 0. \end{cases}$$

This closely matches the target function $f^\star$.

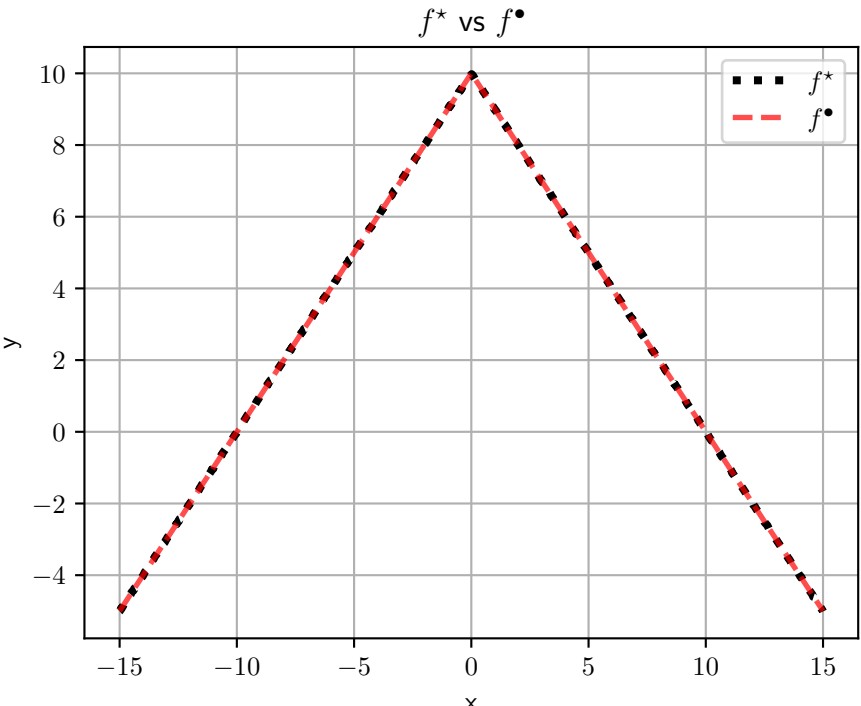

*Figure 16.* Original function vs. best candidate model

One may ask whether the two incoming and outgoing edges of the ReLU node are necessary. To answer this, we can easily synthesize this term by inhabiting the intersection type

$\tau' := \mathbf{D}_2(\mathrm{IN}(1) \cap \mathrm{OUT}(1) \cap$

$\quad \mathrm{STRUCT}([((\mathrm{Copy}, 3), 1, 3)] \bigotimes [((\mathrm{ReLU}, ()), 1, 1) \bigotimes ((\mathrm{Linear}, (2, 1, \mathrm{True})), 2, 1)] \bigotimes [((\mathrm{Linear}, (3, 1, \mathrm{True})), 3, 1)]))$

and apply $f_{\mathrm{obj}}$ to it. The $f_{\mathrm{obj}}$-value is basically the same. Therefore, during our experiments, we didn't just find a near-optimal term for $f_{\mathrm{obj}}$, but also one with a near-minimal amount of parameters.

This example illustrates how the synthesized search spaces allow the discovery of compact and interpretable parametric functions that capture the underlying compositional structure of the target process.

