# OpenReview forum: "Search Space Synthesis for Parametric Functions"
_ICML.cc/2026/Conference — ICML 2026 regular_

### Official Review · Reviewer_86nj · 2026-03-08

**Soundness:** 2
**Presentation:** 2
**Significance:** 3
**Originality:** 3
**Overall Recommendation:** 3
**Confidence:** 3

**Summary:**

This paper is about building the search space automatically, instead of making the user hand-design a NAS or HPO search grammar. The authors model ML components as parametric functions, encode them in typed combinatory logic, and use CoSy to synthesize valid compositions from a specification. They then extend this setup so the synthesized solution spaces can actually be searched using sampling, evolutionary search, and Bayesian optimization. The prototype is implemented around PyTorch like components, and the paper argues that both NAS and HPO can be expressed inside this framework.

**Compliance With Llm Reviewing Policy:**

Affirmed.

**Key Questions For Authors:**

What do you want the reader to treat as the main contribution?

Can you show a stronger empirical comparison against existing grammar-based NAS or DSL-based search approaches?

How should readers interpret the statement that there is no categorical model for FCLP?

How often do the undecidability / nontermination issues become a real bottleneck in practice?

**Limitations:**

yes

**Strengths And Weaknesses:**

I think the paper has a real idea in it. The idea to generate the search space from typed component specifications, instead of fixing the search space first and only searching inside it is interesting and, at least in this form, fairly unusual. The paper also does a decent job explaining how it differs from grammar-based NAS work: the point is not just another grammar, but automatic synthesis of admissible spaces from component specifications, with typing used to guarantee well-formedness. That part is the paper’s clearest contribution.

The paper gives an algebraic presentation of parametric functions using string-diagram constructors and equations, states that this yields a Para-construction, then explains how to encode the resulting objects in FCLP and how to use CoSy to compute solution spaces. It also extends those solution spaces with sampling and search procedures and discusses two search methods built on top: evolutionary search and Bayesian optimization.

The implementation is presented as a proof of concept, and the current experimental section is most convincing in that role. The PyTorch example helps make the framework concrete, and the proposed practical advantages are sensible. However, the evaluation is still fairly limited, and I would have liked at least one benchmark comparison against existing grammar-based NAS/HPO approaches on search quality, cost, or scalability.

The structure is logical, but the paper is dense, and the main narrative never really settles at one level of abstraction. The authors say they leave details to the appendix “to keep the topic accessible,” but the main paper still feels quite abstract.

---

> ### Author Rebuttal · Authors · 2026-03-30
>
> We sincerely thank Reviewer 86nj for the encouraging review. In the following, we will address the provided questions.
>
> ## What do you want the reader to treat as the main contribution? ##
>
> The main contribution is the type environment in Figure 9, which encodes the categorical model for parametric functions in FCLP.
>
> To our knowledge, it is a novel contribution to apply category theory to topics like neural architecture search.
>
> This results in a request language for search spaces of parametric functions that allow the user to ask for specific architectural variance, as well as parameter variance controlled by the choice of the literal repository Delta.
> Additionally, the way the parametric intersection types are constructed ensures that the inhabitation algorithm computes finite search spaces that behave favourably under the provided enumeration and searching methods.
> As you pointed out, the result is relatively abstract. But we believe this to be a feature, as the type environment in Figure 9 can easily be adapted to different problem domains for parametric functions.
> In our proof-of-concept implementation, we chose to interpret the synthesized terms as PyTorch modules to showcase their applicability within a common framework, but this is just one possible domain of interpretation, thanks to the language-agnostic nature of the CoSy framework.
>
> ## Can you show a stronger empirical comparison against existing grammar-based NAS or DSL-based search approaches? ##
>
> The existing grammar-based approaches mainly focus on how to search a specific search space. As discussed in the above question, the paper focuses on synthesizing the search space and less on how to search it.
>
> Providing a sampling mechanism, genetic operators for evolutionary algorithms, and a procedure for Bayesian Optimization enables an empirical comparison for a fixed search space, but wouldn't really fit the focus of the paper.
>
> Therefore, we would claim that a strong empirical comparison against existing approaches is difficult due to the different levels of abstraction and the novelty of our contribution.
>
> What we could provide are some numbers from our proof of concept implementation, but the use case doesn't really match the existing approaches.
> Are there any performance metrics we could provide from the proof of concept implementation that would strengthen our results in your opinion?
> Then we will gladly provide them in the camera-ready version of the paper, utilizing the additional page.
>
>
> ## How should readers interpret the statement that there is no categorical model for FCLP? ##
>
> From a formal perspective, we have shown how to encode parametric functions as morphisms in the category of parametric functions in FCLP.
> Proving the correctness of this encoding would be the next step. But this is difficult, as to our knowledge, there is no known categorical semantics for FCLP, or even combinatory logic with intersection types. To get an idea, why this is a non-trivial problem: Take a combinator C typed with an intersection type a & (a -> a).
> This combinator can be assigned the type a as well as the type a -> a, and therefore, it is a function that has multiple arities.
> When trying to come up with a categorical model for combinatory logic with intersection types, this is one of the earliest hurdles one encounters.
>
> One reason we weren't able to provide a correctness proof for our encoding is the absence of a categorical model for FCLP.
>
> ## How often do the undecidability / nontermination issues become a real bottleneck in practice? ##
>
> The undecidability and semi-decidability of type inhabitation in FCLP rarely become an issue in practice. Otherwise, the CoSy framework wouldn't be used successfully for software synthesis.
> Our extension of this framework doesn't change the inhabitation algorithm. It conservatively extends the synthesized tree structure with additional fields for information required for the more efficient implementation of the genetic algorithms.
> And we provide a sampling method, which is also a conservative extension, as there was just a (bottom-up) enumeration of terms implemented in CoSy.
> But of course, one can come up with pathological examples for type environments, where nontermination is unavoidable. Therefore, an honest answer would be: It depends on how you model your problem as a type environment.
> The model provided in section 4 and the discussion on how and why we came up with the concrete types are mostly to have a good performance.
> So if a user uses the type environment from Figure 9 as a template and only selects appropriate functions and parameter spaces, they won't encounter nontermination.
> Especially, because the provided model can only produce finite search spaces.
> You can change it to also describe possibly infinite search spaces, then nontermination can become an issue for certain requests.
> That is one reason why we chose to enforce finiteness.

---

> > ### Author Rebuttal · Reviewer_86nj · 2026-04-05
> >
> > Thank you for the rebuttal. The clarification of the intended contribution and the discussion of the FCLP/nontermination issue are helpful, but my main concern about limited empirical validation relative to existing grammar-based NAS/HPO approaches remains. I therefore view my concerns as only partially resolved and do not plan to change my score.

---

> > > ### Author Response · Authors · 2026-04-07
> > >
> > > Dear Reviewer 86nj,
> > >
> > > Thank you for your time and effort. We are sorry to hear that our provided clarification does not resolve your issue.
> > > In our above clarification, we explicitly asked you what kind of empirical validation you want us to do, as we would be encouraged by such constructive feedback to provide the empirical results you claim are missing.
> > > Nevertheless, we appreciate your feedback and are thankful for your valuable input.

---

### Official Review · Reviewer_zhAV · 2026-03-11

**Soundness:** 3
**Presentation:** 3
**Significance:** 3
**Originality:** 3
**Overall Recommendation:** 4
**Confidence:** 1

**Summary:**

This paper presents a general framework for synthesizing the search space of parametric functions, along with strategies for traversing these spaces to discover optimal solutions. The authors develop an algebraic theory formalization of a categorical model for
  parametric functions within Finite Combinatory Logic with Predicates (FCLP). Building upon this FCLP-based component-oriented synthesis framework, the approach enables automatic composition and search of parametric functions from given components.

**Compliance With Llm Reviewing Policy:**

Affirmed.

**Key Questions For Authors:**

None

**Limitations:**

yes

**Strengths And Weaknesses:**

This paper is purely theoretical, and I am not an expert in this area. I am unsure why it was assigned to me. I regret that, as a reviewer, my expertise is limited, and I am unable to provide an in-depth review or assess the validity of the theory. However, at first glance, the overall quality of the paper appears to be good. The potential issue is that I do not fully grasp the significance of the theory or its connection to machine learning; it remains unclear how the theory could be applied to current practical machine learning. I give a weak accept with the lowest confidence, and I suggest AC to place greater weight on the opinions of other reviewers.

---

> ### Author Rebuttal · Authors · 2026-03-30
>
> We sincerely thank Reviewer zhAV for their review.
> You raised the question of how the presented theoretical results can be applied to current machine learning, and we would like to point you to our answer for Reviewer 86nj question "What do you want the reader to treat as the main contribution?" as it should answer your question as well.

---

> > ### Author Rebuttal · Reviewer_zhAV · 2026-04-04
> >
> > I stand by my opinion.

---

> > > ### Author Response · Authors · 2026-04-07
> > >
> > > Dear Reviewer zhAV,
> > >
> > > Thank you for your time and effort.

---

### Official Review · Reviewer_csps · 2026-03-24

**Soundness:** 3
**Presentation:** 3
**Significance:** 2
**Originality:** 2
**Overall Recommendation:** 3
**Confidence:** 3

**Summary:**

The paper presents a method to synthesize search spaces of parametric functions for choosing components in an ML task, and an optimization approach to find optimal components in these spaces.

**Compliance With Llm Reviewing Policy:**

Affirmed.

**Key Questions For Authors:**

How are the subcomponents chosen for sampling and "¿"-masking in the optimization?

In which problems have the advantages listed in Chapter 5 been observed and in comparison to which alternative methods?

Does the optimization approach assume that each component can be optimized independently? How does the approach and the parallel evaluation change if parameters have a strong influence on each other?

In which settings is the "larger" search space proposed in this work most beneficial? How do constraints, that make certain components more useful in certain parts of the pipeline than others, influence the search space and optimization?

**Limitations:**

Some discussion in Chapter 2.

The authors' impact statement is adequate.

**Strengths And Weaknesses:**

**Soundness:**
The method to generate the search space is sound and clearly explained.
I did not check their proofs.

**Presentation**
The paper is clearly written and presents a detailed background to the topic. Illustrations and examples are well-chosen.

**Significance**

Providing a larger search space that is easier to define is useful for many optimization tasks.

A weakness is the reproducibility of their optimization approaches.
The description of their learning approach is very abstract and lacks details on the evolutionary sampling, Bayesian approach and optimization.

**Originality**
The approach to generate the search space seems new. The usage of their approach in an optimization is very abstract.

**Typos**
p1 "BOHNAS (Schrodi et al., 2023),and Einspace" space missing.
p2 "Consider affine transformations as an example": Is this supposed to be a "linear transformation"?
p4 "definite-clause grammars(Wong & Leung" space missing.
p6 "Figure 9will" space missing.
p7 "Figure 9. Caption".

---

> ### Author Rebuttal · Authors · 2026-03-30
>
> We sincerely thank Reviewer csps for their comments and suggestions. Below, we address the question raised.
>
> ## How are the subcomponents chosen for sampling and "¿"-masking in the optimization? ##
>
> The "¿"-masking is a feature of the request language to precisely describe the architectural variance within a search space.
> Variance means all subcomponents that can be type-correctly inserted at this point.
> Please note that it is a literal in a parametric intersection type.
> For a given request (parametric intersection type), the CoSy framework computes a parameterized tree grammar (search space) describing the set of inhabitants of the request.
> This parameterized tree grammar then describes all combinatorial terms (inhabitants) in which the subcomponents have been varied at the points marked with the "¿"-mask.
> Therefore, the parameterized tree grammar describes the variance of all subcomponents that can be chosen for sampling and optimization, since the optimization is closed under it.
>
> ## In which problems have the advantages listed in Chapter 5 been observed and in comparison to which alternative methods? ##
>
> The advantages listed in Chapter 5 are novel functionalities enabled by search space synthesis. We observed them evaluating the prototypical implementation of the proof of concept presented.
> Therefore, it is difficult to compare them with alternative methods, as we are not aware of any.
> Do you have any alternative method in mind for which a comparison would strengthen our contribution?
>
> ## Does the optimization approach assume that each component can be optimized independently? How does the approach and the parallel evaluation change if parameters have a strong influence on each other? ##
>
> No, the components are not optimised independently. The search space is intuitively the product of all possible architectures and the possible parameter values for each component, and is therefore fairly large (but still finite). That's why it's so important to be able to request specific search spaces in which only the requested architectural substructures and the parameters for their components vary. Or to put it the other way, the presented request languages enable a user to fix certain parts of an architecture as well as the parameters of its components and request a search space that describes only variance in the substructures marked by "¿".
> The type environment shown in Figure 9 enables a user to define requests for disjunct search spaces, for example, by ensuring that the sizes of each architecture in one search space are smaller than those of every architecture in another.
> We assume that by parallel evaluation, you refer to the point listed in section 5, where one is able to manually parallelise the search. The idea here is that our presented approach enables a user to precisely describe the architectural variance as well as the variance of the parameters for the components of such architectures. Therefore, a big search space can be partitioned into smaller search spaces. Keep in mind that a search space is a parameterized tree grammar synthesized from a requested parametric intersection type and optimization means searching for the best combinatorial term in the language of this grammar under a given scoring. Therefore, each partition can be searched on its own and describes a smaller search problem than searching a (much) bigger search space.
> The strong influence that parameters may have on each other doesn't change anything here, and thank you for pointing this out. Maybe we should try to highlight this, to make the significance of our contribution clearer.
>
> ## In which settings is the "larger" search space proposed in this work most beneficial? How do constraints, that make certain components more useful in certain parts of the pipeline than others, influence the search space and optimization? ##
>
> Our contribution is not proposing a larger search space, but providing an abstract encoding of search spaces of parametric functions inspired by the categorical model.
> As the resulting request language enables the user to precisely describe the architectural variance, one is able to fix certain substructures and only vary the components in the parts where one isn't sure about their usefulness.
> This doesn't influence the search space or the optimization, as this is exactly the contribution of the presented approach to synthesize search spaces from a request describing the variance of substructures.

---

> > ### Author Rebuttal · Reviewer_csps · 2026-04-05
> >
> > Thank you for your reply! While describing the search space with a grammar can be advantageous, it is still unclear to me how the search space is used effectively within the sampling or optimization setting. Clarifying this would make it easier to appreciate the flexibility of querying specific search spaces.
> > I maintain my score.

---

> > > ### Author Response · Authors · 2026-04-07
> > >
> > > Dear Reviewer csps,
> > >
> > > Thank you for your time and feedback.
> > > We will try to clarify your remaining question as follows:
> > > The focus of the paper is on synthesizing search spaces for parametric functions, as we strongly believe that the combination of combinatory logic and category theory for this use case is a novel and significant contribution.
> > > The CoSy framework encodes the synthesis result as a parameterized tree grammar, and therefore, we need to reason that this structure can be searched. Our results aren't any technically new methods for sampling or grammar-guided genetic programming to search these structures, but existing methods for genetic programming closed under definite clause grammars and sampling of context-free grammars can be adapted to parameterized tree grammars.
> > > Additionally, by providing sampling and genetic programming mechanisms, we show that Bayesian optimization can also be adapted with a few extra steps, as we can apply genetic programming to optimize the acquisition function.
> > > Therefore, we have shown that existing (grammar-based) search/optimization methods can be applied to parameterized tree grammars as well.
> > > We explicitly do not provide new technical insights on searching/optimization, but our main contribution is a method to synthesize search spaces with a strong formal foundation.

---

### Official Review · Reviewer_Quv9 · 2026-03-26

**Soundness:** 4
**Presentation:** 4
**Significance:** 4
**Originality:** 4
**Overall Recommendation:** 6
**Confidence:** 3

**Summary:**

This paper introduces a mathematical system for searching through the space of parametric functions, similar to neural architecture search but much more general and rigorous. It offers improvements over similar systems

**Compliance With Llm Reviewing Policy:**

Affirmed.

**Final Justification:**

This is a strong theoretical paper providing a general framework for searching through parameterized functions. I'd love to see a practical extension, but the work is strong as is and all of my questions/concerns have been addressed.

**Key Questions For Authors:**

I will preface by saying that I am not deeply familiar with category theory (though I would like to be), so please understand my questions may be based on incorrect assumptions/misunderstandings. My closest familiarity is with working with architecture search systems like AutoML Zero, and I have good familiarity with program synthesis.

- could the authors provide examples of how to recover common neural network architectures? I understand that this isn't the core focus of the paper, but would've helped me in grounding and understanding it (and likely would for the average ICML reader that isn't a category theorist, e.g. ML practitioners). Beyond improving understanding, it would also better "sell" the paper. e.g. can this search system recover convolutional networks like AutoML Zero can (if it had sufficient computational resources)?

- I do also wonder about how efficient/feasible searching this space actually is? While there may be advantages over grammar-based systems, grammar based systems do allow "biasing" the search space via operators that are more likely to confer advantages when compared to having (in the instance of evolutionary search) crossover and mutation operators in terms of primitives.

- If I've understood correctly, the system focuses on "external" search over parametric functions, and also covers an inner loop for optimizing their parameters? Or is this just treated as the scoring function? (I noticed Fong 2019 is cited)

- How does the framework compare to other systems cited, e.g. einspace

**Limitations:**

Technically the paper does not have a limitations section, but I feel that limitations were presented adequately throughout the paper.

**Strengths And Weaknesses:**

Generally:

- I have not carefully reviewed the math presented, but from my level of understanding, the paper is sound.
- The presentation is appropriate for someone with the appropriate background (though stylistic choices or future work could make it appealing to broader audience. In particular, I understand this paper through the lens of systems like [1], e.g. for neural architecture search. I would've understood the paper better at a practical level if it gave examples of how to recover modern neural network architectures (as a special case of parametric functions), e.g. feedforward, conv, rnn, transformer --- but I understand that the authors were focused on writing a theoretical paper.
- Searching through the space of parametric functions is critical for a lot of machine learning tasks, and having a rigorous/elegant mathematical system to do this in is likely to have high impact, e.g. a best case scenario would be discovering or generalizing particular neural architectures.
- Originality: there is a categorical theory of neural architectures (already cited: "Position: Categorical
deep learning is an algebraic theory of all architectures.") but this differs from a search space over parametric functions. To the best of my knowledge, the work is novel and appropriately cites previous work. However, I'd recommend adding more citations to architecture search, which has a larger body of work (e.g. [1])

- The specific improvement on CoSy for enabling scoring functions seems like a core contribution that would move this system closer to practical applications

[1] Real, E., Liang, C., So, D., & Le, Q. (2020, November). Automl-zero: Evolving machine learning algorithms from scratch. In International conference on machine learning (pp. 8007-8019). Pmlr.

---

> ### Author Rebuttal · Authors · 2026-03-30
>
> We sincerely thank Reviewer Quv9 for the positive review and the encouraging feedback. Could you elaborate on why you judged the soundness of our contribution to be poor, or was it simply a misclicked input?
> We address the questions raised below.
>
> Questions:
> ## Could the authors provide examples of how to recover common neural network architectures? I understand that this isn't the core focus of the paper, but it would've helped me in grounding and understanding it (and likely would for the average ICML reader that isn't a category theorist, e.g., ML practitioners). Beyond improving understanding, it would also better "sell" the paper. e.g., can this search system recover convolutional networks like AutoML Zero can (if it had sufficient computational resources)? ##
>
> Thank you for pointing this out. We will gladly use the additional page for the camera-ready version to provide a more detailed description of how our proof-of-concept implementation recovers neural network architectures.
>
> Synthesizing search spaces for CNNs is also possible. The type environment shown in Figure 9 describes, intuitively speaking, the synthesis of parametric functions of type R^n x P -> R^m, where P is a suitable parameter space.
> Of course, one could apply this for synthesizing CNNs, but it would currently require the user to compute suitable dimensions n and m from the kernel parameters given in P. Therefore, we would suggest some adaptations to the type environment in Figure 9 for CNNs. We thought this to be out of scope, as it is more of a specialisation of the presented approach than a generalisation, but we could provide an adapted type environment for the interested reader in the appendix.
>
>
>
> ## I do also wonder about how efficient/feasible searching this space actually is? While there may be advantages over grammar-based systems, grammar based systems do allow "biasing" the search space via operators that are more likely to confer advantages when compared to having (in the instance of evolutionary search) crossover and mutation operators in terms of primitives. ##
>
> We are sorry, but we don't fully understand this question. We will try to come up with a hopefully satisfactory answer:
> The produced search space is a parameterized tree grammar. Being a tree grammar has the advantage that it treats trees as elements of the grammar rather than words/strings. Therefore, the genetic algorithms don't need to be defined on the derivation tree of a word of the language of a grammar, but can work directly on the elements.
> But this is still a grammar-based approach; the key difference is that it works with tree languages rather than word languages.
>
> So the "biasing" of the search space can be achieved directly by asking the synthesis to generate the "biased" search space by formulating a corresponding request, right?
> Because every synthesized search space is closed under the provided search methods by construction.
>
> Or do you mean approaches like probabilistic grammars? This is a topic we are currently researching. The idea here would be that a parameterized tree grammar is equivalent to a logic program in a certain amalgamation of Horn Logic. Therefore, what you call "biasing" may be equivalent to Probabilistic Inductive Logic Programming. This approach is currently future work and out of scope for this paper. But this is on our stack for future work.
>
>
>
> ## If I've understood correctly, the system focuses on "external" search over parametric functions, and also covers an inner loop for optimizing their parameters? Or is this just treated as the scoring function? (I noticed Fong 2019 is cited) ##
>
> No, there is no inner loop for parameter optimisation. The search space is intuitively the product of all possible architectures and the possible parameter values for each component, and is therefore fairly large (but still finite). That's why it's so important to be able to request specific search spaces in which only the requested architectural substructures and their component parameters vary. Or, to put it another way, the presented request languages enable a user to fix certain parts of an architecture, adjust the parameters of its components, and request a search space that describes only variance in the substructures marked by "¿".
> The scoring maps elements of the search space to a score and therefore can't optimise the parameters, but evaluates an architecture and its components with fixed parameters. The search space describes the architectural and parameter variances of the components specified in the synthesis request.
>
>
> ## How does the framework compare to other systems cited, e.g. einspace? ##
>
> We would like to point you to our answers for Reviewer 86nj, as they should answer this question as well.

---

### Decision · Program_Chairs · 2026-04-30

**Decision:**

Accept (regular)

**Comment:**

This work proposes a framework for building search spaces of parametric functions from typed components using a categorical formulation. These spaces are intended to work with standard black-box optimization methods such as evolutionary algorithms and Bayesian optimization.

The reviews are mixed. Reviewer Quv9 is strongly positive, highlighting the novelty and potential impact, and notes that the rebuttal addressed their concerns. Reviewer zhAV is positive but remains uncertain, describing the approach as technically sound while questioning its practical relevance. Reviewer csps gives a weak reject, acknowledging the soundness of the search space construction but pointing out that the optimization component is not developed enough to assess practical usefulness. Reviewer 86nj also assigns a weak reject, emphasizing the lack of empirical validation and comparisons to related approaches like neural architecture search or grammar-based methods.

There is overall agreement that the idea of generating search spaces from typed specifications is novel and technically sound. The main point of disagreement is how well the paper demonstrates practical value. In particular, the discussion on using evolutionary search, and Bayesian optimization remains quite abstract, and the paper provides limited empirical evidence showing that these search spaces are effective in realistic settings. The rebuttal helps clarify that the primary goal is search space construction rather than new optimization methods, which improves the positioning of the work, but it does not fully resolve concerns about the lack of empirical support.

Overall, this is a promising idea that is not yet fully developed. The framework is conceptually strong and could have impact, but the current version falls short in clearly demonstrating its practical effectiveness. I recommend a weak accept, with the expectation that the final version improves clarity and provides more concrete evidence of practical relevance.